# Acute melanization of silkworm hemolymph by peptidoglycans of the human commensal bacterium *Cutibacterium acnes*

**Yasuhiko Matsumoto●\*, Eri Sato, Takashi Sugita**

Department of Microbiology, Meiji Pharmaceutical University, Kiyose, Tokyo, Japan

\* ymatsumoto@my-pharm.ac.jp

## Abstract

*Cutibacterium acnes* is a pathogenic bacterium that cause inflammatory diseases of the skin and intervertebral discs. The immune activation induced by *C. acnes* requires multiple cellular responses in the host. Silkworm, an invertebrate, generates melanin by phenoloxidase upon recognizing bacterial or fungal components. Therefore, the melanization reaction can be used as an indicator of innate immune activation. A silkworm infection model was developed for evaluating the virulence of *C. acnes*, but a system for evaluating the induction of innate immunity by *C. acnes* using melanization as an indicator has not yet been established. Here we demonstrated that *C. acnes* rapidly causes melanization of the silkworm hemolymph. On the other hand, *Staphylococcus aureus*, a gram-positive bacterium identical to *C. acnes*, does not cause immediate melanization. Even injection of heat-killed *C. acnes* cells caused melanization of the silkworm hemolymph. DNase, RNase, and protease treatment of the heat-treated *C. acnes* cells did not decrease the silkworm hemolymph melanization. Treatment with peptidoglycan-degrading enzymes, such as lysostaphin and lysozyme, however, decreased the induction of melanization by the heat-treated *C. acnes* cells. These findings suggest that silkworm hemolymph melanization may be a useful indicator to evaluate innate immune activation by *C. acnes* and that *C. acnes* peptidoglycans are involved in the induction of innate immunity in silkworms.

**Data Availability Statement:** All relevant data are within the article and its Supporting Information files.

## Introduction

*Cutibacterium acnes*, a human commensal bacterium, causes inflammatory skin diseases such as acne vulgaris and inflammation in intervertebral discs [1–3]. Acne vulgaris is inflammation of the hair follicles and sebaceous glands that results in the formation of comedones [3]. *C. acnes* was detected in 34% to 36.2% of intervertebral discs removed from patients with chronic back pain due to local inflammation such as herniated discs [4, 5]. Therefore, understanding the mechanisms underlying the induction of host immunity by *C. acnes* may contribute to preventing and treating those diseases.

*C. acnes* secretes proteins such as lipases, which cause inflammation [6], or directly interacts with Toll-like receptor (TLR) 2 and TLR4 on keratinocytes and immune cells to cause

**Funding:** This study was supported by Kose Cosmetology Research Foundation (No. 711 to Y. M.), JSPS KAKENHI grant number JP20K07022 (Scientific Research (C) to Y.M.), and in part by Research Program on Emerging and Re-emerging Infectious Diseases of the Japan Agency for Medical Research and Development, AMED (Grant number JP22wm0325054 to Y.M.). The funders had no role in study design, data collection and analysis, decision to publish, or preparation of the manuscript.

**Competing interests:** The authors have declared that no competing interests exist.

inflammation [7, 8]. Activation of the innate immune system by *C. acnes* is the cause of the inflammation [9–13]. Immune activation induced by *C. acnes* requires multiple cellular responses in the host, and systemic *C. acnes* infection models have been established in mammals such as mice and rats to evaluate the mechanisms of the chronic inflammation [14, 15]. Long-term mammalian infection experiments using a large number of individuals, however, are problematic from the viewpoint of animal ethics [16].

Silkworms are useful laboratory animals for evaluating the pathogenic mechanisms of microorganisms that cause systemic infections and for assessing the activation of innate immunity [16–19]. Silkworms have several advantages in experiments using large numbers of individuals, and fewer ethical issues are associated with their use [20]. Blood sampling and quantitative drug administration are easy to perform in silkworms, and biochemical parameters in silkworm hemolymph can be determined [21–24].

One innate immune response in insects, including silkworms, is hemolymph melanization [25, 26]. In *Drosophila melanogaster*, hemolymph melanization caused by the recognition of a foreign invader such as bacteria or fungi in the body contributes to coagulating the invader and repairing the wound [25, 26]. The hemolymph melanization and Toll pathway-dependent immune responses are mediated by the same recognition steps through pattern recognition receptors and cofactors [25–27]. Hemolymph melanization was caused by phenoloxidases activated by serine proteases [25, 26]. The phenoloxidases (PPO1 and PPO2)-deficient flies are sensitive to *Staphylococcus aureus* infection [28]. Therefore, hemolymph melanization in insects can be used as an indicator of innate immune system activation [17, 19, 29]. The induction of innate immunity by *Porphyromonas gingivalis* and *Candida albicans* was evaluated by silkworm hemolymph melanization [24, 30]. A silkworm infection model with *C. acnes* was also established for evaluating antimicrobial drug efficacy [31]. A system for evaluating the induction of innate immunity by *C. acnes* on the basis of silkworm hemolymph melanization, however, has not yet been developed.

In the present study, we established a system for evaluating the induction of innate immunity by *C. acnes* using silkworm hemolymph melanization as an indicator and show that water-insoluble *C. acnes* peptidoglycans are involved in inducing innate immunity.

## Materials & methods

### Reagents

Gifu anaerobic medium agar was purchased from Nissui Pharmaceutical Co., Ltd. (Tokyo, Japan). Tryptic soy broth was purchased from Becton Dickinson (Franklin Lakes, NJ, USA). Protease K was purchased from QIAGEN (Hilden, Germany). RNase A was purchased from NIPPON GENE, Co., Ltd. (Tokyo, Japan). DNase was purchased from Promega Corporation (WI, USA). Lysostaphin, lysozyme, methanol, and chloroform were purchased from FUJI-FILM Wako Pure Chemical Corporation (Osaka, Japan). *Staphylococcus aureus* peptidoglycans, *Bacillus subtilis* peptidoglycans, and *Micrococcus luteus* peptidoglycans were purchased from Sigma-Aldrich (St. Louis, MO, USA).

### Culture of bacteria

The *C. acnes* ATCC6919 strain and *Staphylococcus aureus* Newman strain were used in this study. The *C. acnes* ATCC6919 strain was spread on Gifu anaerobic medium agar and incubated under anaerobic conditions at 37˚C for 3 days [31]. *S. aureus* Newman strains were spread on tryptic soy broth agar and incubated under aerobic conditions at 37˚C for 1 day.

## Silkworm rearing

Silkworm rearing procedures were described previously [32]. Silkworm eggs were purchased from Ehime-Sanshu Co., Ltd. (Ehime, Japan), disinfected, and hatched at 25–27 ˚C. The silkworms were fed an artificial diet, Silkmate 2S, containing antibiotics purchased from Ehime-Sanshu Co., Ltd. Fifth instar larvae were used in the infection experiments. The silkworm infection experiments were performed as previously described [32]. Silkworm fifth instar larvae were fed an artificial diet (1.5 g; Silkmate 2S; Ehime-Sanshu Co., Ltd) overnight. A 50-μl suspension of *C. acnes* cells was injected into the silkworm hemolymph with a 1-ml tuberculin syringe (Terumo Medical Corporation, Tokyo, Japan). Silkworms injected with the *C. acnes* cells were placed in an incubator and survival was monitored.

## *In vivo* melanization assay

An *in vivo* melanization assay was performed as previously described [30] with slight modification. Hemolymph was collected from the larvae through a cut on the first proleg as described previously [33]. The silkworm hemolymph (50 μl) was mixed with 50 μL of physiologic saline solution (0.9% NaCl: PSS). Absorbance at 490 nm was measured using a microplate reader (iMark™ microplate reader; Bio-Rad Laboratories Inc., Hercules, CA, USA). The arbitrary unit was defined as the absorbance at 490 nm of a sample of silkworm hemolymph (50 μl) mixed with saline (50 μl).

## DNase, RNase, and protease treatment

Autoclaved *C. acnes* cells (AC) were diluted with phosphate buffered saline (PBS) to absorbance at 600 nm ($A_{600}$) = 3 in 1 mL, and 10 μL each of DNase (1 U/μl) and RNase (100 mg/ml) was added. After incubation for 2 h at 37˚C, the suspension was centrifuged at 15,000 rpm for 15 min at room temperature. The precipitate was suspended with 500 μL of PSS, diluted with PBS to $A_{600}$ = 2 in 1 mL, and 50 μL of protease (0.75 AU/ml) was added. After incubation for 1 h at 50˚C, the sample was centrifuged at 15,000 rpm for 10 min at room temperature. The precipitate was suspended with 1 mL of PSS, and the remaining enzymes were inactivated by incubation at 80˚C for 30 min. The samples were centrifuged at 15,000 rpm for 10 min at room temperature, and the precipitate was diluted with PSS to $A_{600}$ = 1 to make the AC-En sample.

## Bligh-Dyer method

The AC-En sample (1 mL) was mixed with 1 mL of distilled water, 2.5 mL of methanol, and 1.25 mL of chloroform by shaking for 2 min. The sample was allowed to stand at room temperature for 10 min; 1.25 mL of chloroform was then added and the mixture shaken vigorously for 30 s. Next, 1.25 mL of distilled water was added, and the sample was mixed vigorously for 30 s and centrifuged at 3,000 rpm for 5 min at room temperature. The aqueous fraction was collected by removing the organic solvent layer. The extracted aqueous fraction was further centrifuged at 15,000 rpm for 15 min at room temperature and separated into supernatant and precipitate. The precipitate fraction was prepared by adding the same volume of PSS as the supernatant fraction.

## Lysostaphin and/or lysozyme treatment

Lysostaphin (10 mg/ml; 2.5 μL) and lysozyme (100 mg/ml; 5 μL) were added alone or in combination to 300 μL of the precipitate fraction obtained by the Bligh-Dyer method, adjusted to 500 μL using PBS, and the samples were incubated at 37˚C for 24 h. The incubated samples

were centrifuged at 15,000 rpm for 15 min at room temperature and separated into supernatant and precipitate. The precipitate fraction was prepared by adding the same volume of PSS as the supernatant fraction.

## Statistical analysis

Statistical differences between groups were analyzed by the Tukey test or Tukey-Kramer test. Each experiment was performed at least twice and error bars indicate the standard deviations of the means. A P value of less than 0.05 was considered statistically significant.

## Results

### Silkworm hemolymph melanization caused by the injection of *C. acnes*

Injection of *C. acnes* cells into silkworm hemolymph killed the silkworms [31]. Based on the silkworm infection model, we examined whether injection of *C. acnes* cells causes melanization, a darkening of the silkworm hemolymph. Silkworm hemolymph melanization was induced by injection of *C. acnes* cells within 3 h (Fig 1). On the other hand, inoculation with *S. aureus*, a human skin commensal gram-positive bacterium like *C. acnes*, did not induce melanization of the silkworm hemolymph at 3 h (Fig 2). These findings suggest that silkworm hemolymph melanization is induced more rapidly by *C. acnes* than by *S. aureus*.

### Melanization of silkworm hemolymph by heat-killed *C. acnes* cells

We next examined whether *C. acnes* survival is important for inducing melanization of the silkworm hemolymph. Heat-killed *C. acnes* cells (*C. acnes* [AC]) were obtained by autoclaving. Melanization-inducing activity of *C. acnes* cells was detected even after the cells were autoclaved (Fig 3A and 3B). Silkworm hemolymph melanization was induced by the heat-killed *C. acnes* cells in a dose-dependent manner (Fig 3C and 3D). These findings suggest that a heat-tolerant bacterial component of *C. acnes* is involved in inducing melanization of the silkworm hemolymph.

### Induction of silkworm hemolymph melanization by water-insoluble *C. acnes* peptidoglycans

We next examined which components of the heat-killed *C. acnes* cells induced melanization of the silkworm hemolymph. Melanization of the silkworm hemolymph was caused by the DNase-, RNase-, and protease-treated fraction (*C. acnes* [AC-En] fraction) of the heat-killed *C. acnes* cells (Fig 4). This result suggests that active substances other than DNA, RNA, and protein in the heat-killed *C. acnes* induce melanization of the silkworm hemolymph. Next, we examined whether the lipid-eliminated fraction obtained by the Bligh-Dyer method from the *C. acnes* (AC-En) fraction exhibited melanization activity. When the *C. acnes* (AC-En) fraction was treated with the Bligh-Dyer procedure, water-soluble and water-insoluble fractions were obtained in addition to the chloroform-methanol fraction (Fig 5A). The water-insoluble fraction (Ppt) induced silkworm hemolymph melanization, but the water-soluble fraction (Sup) did not (Fig 5B and 5C). These findings suggest that the *C. acnes* induces silkworm hemolymph melanization by water-insoluble substances other than DNA, RNA, proteins, and lipids. We hypothesized that peptidoglycans are candidate heat-tolerant water-insoluble substances other than DNA, RNA, proteins, and lipids in *C. acnes*. The water-insoluble fraction (BDppt) was treated with lysostaphin and/or lysozyme, which cleave peptidoglycans (Fig 6A). Moreover, supernatant and precipitate fractions were obtained after treatment with lysostaphin

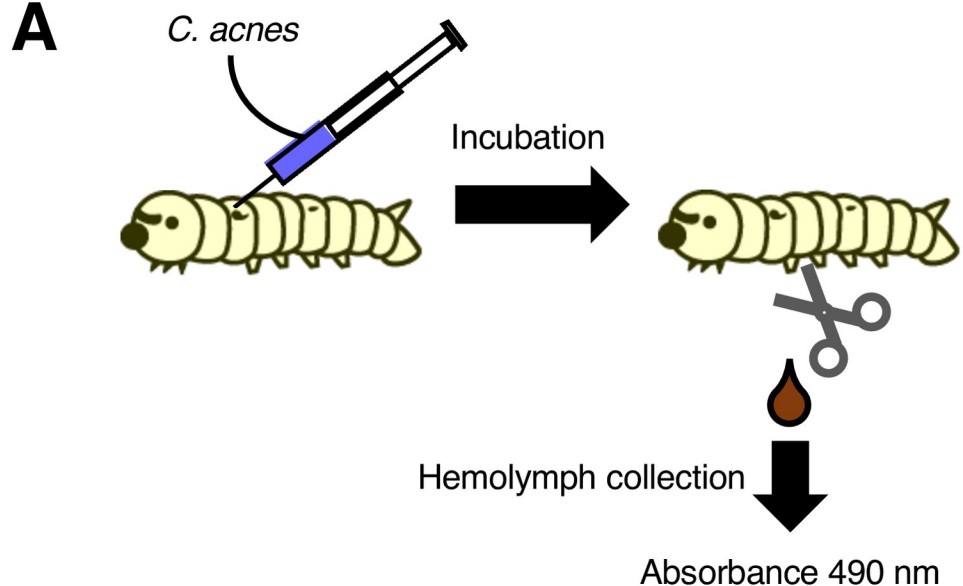

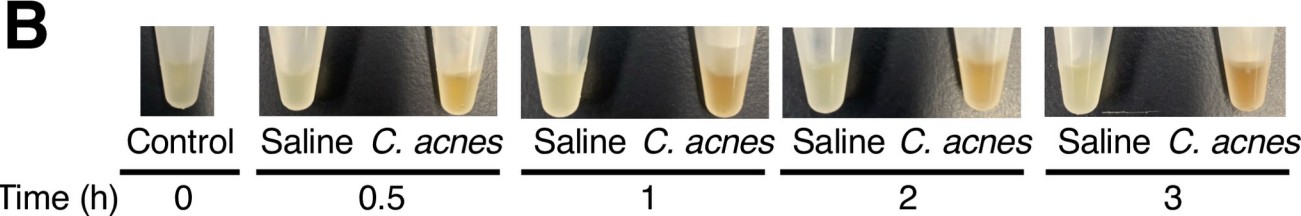

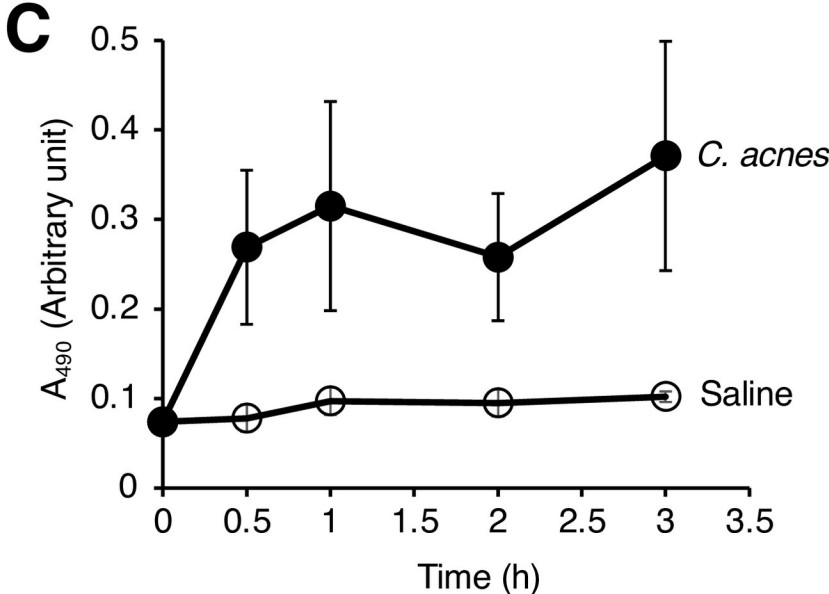

**Fig 1. Induction of silkworm hemolymph melanization by injection of *C. acnes*.** (A) Illustration of an experimental method to determine the silkworm hemolymph melanization. Silkworms were injected with saline (Saline) or *C. acnes* cell suspension (*C. acnes*; 1.1 x 10$^7$ cells/larva), and reared at 37°C. Silkworm hemolymph was collected for 3 h, photographs of the silkworm hemolymph were taken (B), and absorbance at 490 nm (A$_{490}$) (C) was measured. The arbitrary unit was defined as the absorbance at 490 nm of a sample of silkworm hemolymph (50 μl) mixed with saline (50 μl).

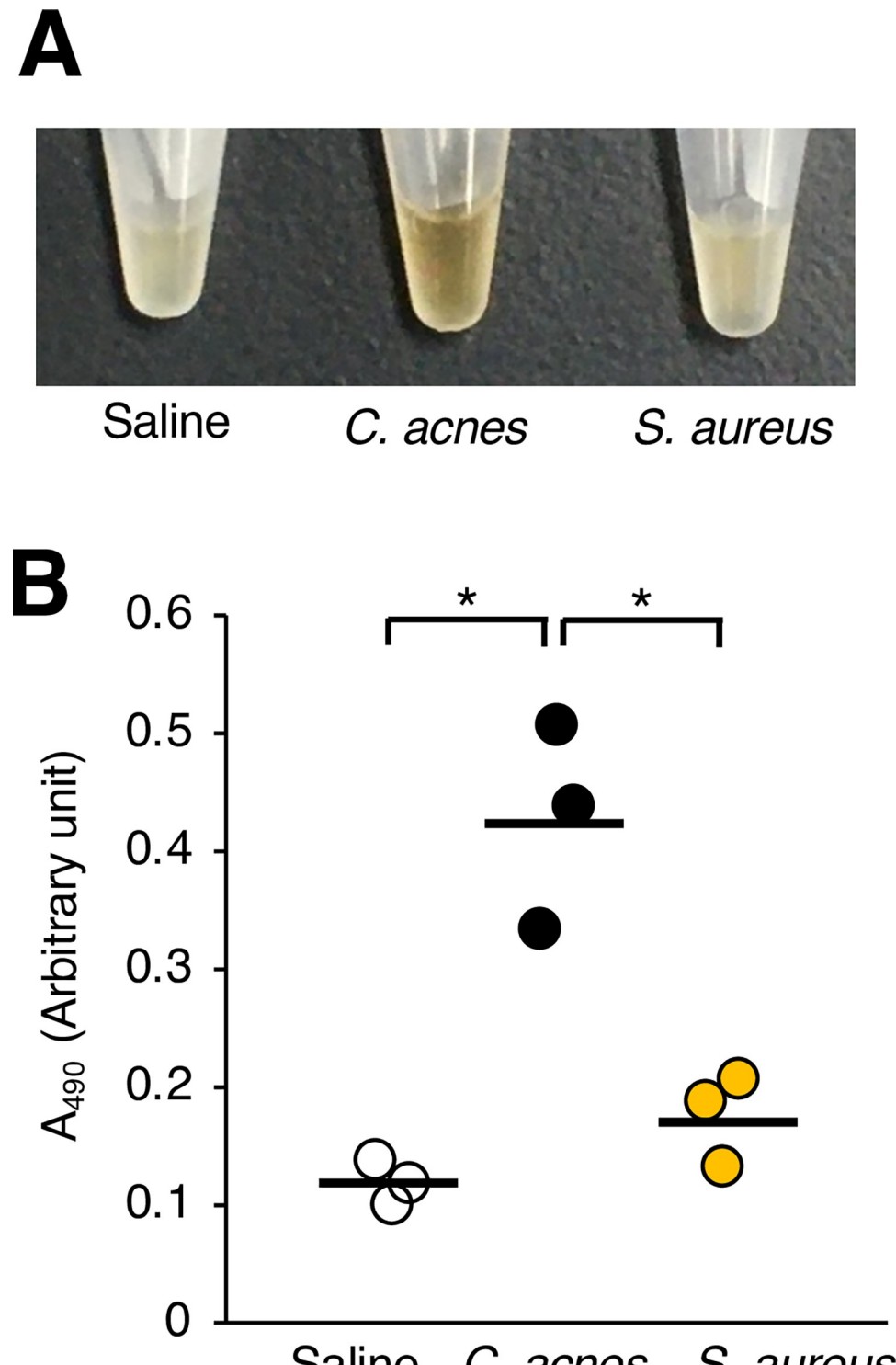

**Fig 2. Comparison of melanization induction by _C. acnes_ and _S. aureus_ in silkworm hemolymph.** Silkworms were injected with saline (Saline), _C. acnes_ cell suspension (_C. acnes_; 1 x $10^8$ cells/larva), or _S. aureus_ cell suspension (_S. aureus_; 1 x $10^8$ cells/larva). After incubation for 3 h at 37°C, the silkworm hemolymph was collected. Photographs of the silkworm hemolymph were taken (**A**), and absorbance at 490 nm ($A_{490}$) (**B**) was measured. The arbitrary unit was defined as the absorbance at 490 nm of a sample of silkworm hemolymph (50 μl) mixed with saline (50 μl). n = 3/ group. Statistically significant differences between groups were evaluated using the Tukey test. *$P < 0.05$.

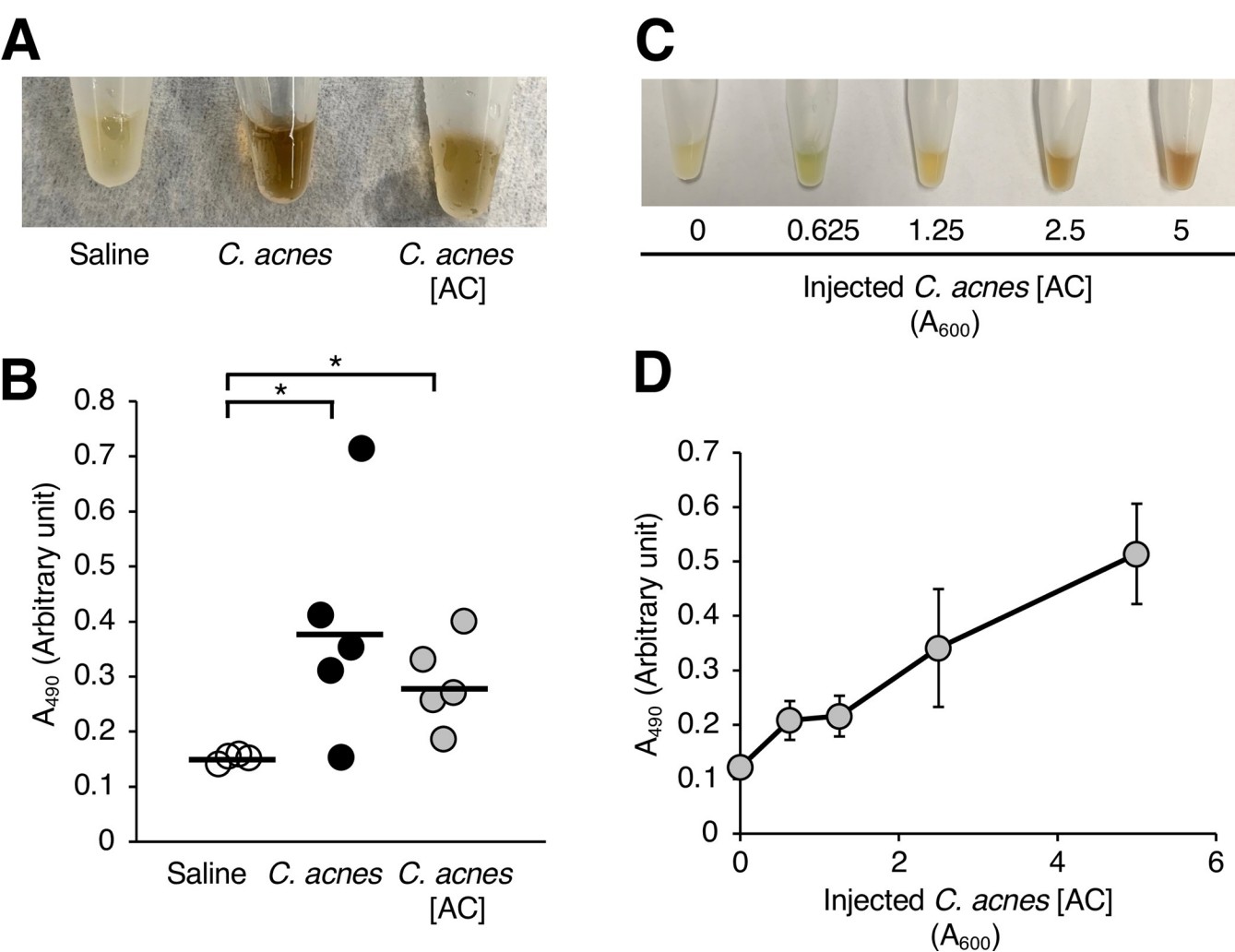

**Fig 3. Melanization of silkworm hemolymph induced by injection of heat–killed *C. acnes* cells.** (**A**, **B**) Silkworms were injected with saline (Saline), *C. acnes* cell suspension (*C. acnes*; 1.1 x 10$^7$ cells/larva), or the autoclaved *C. acnes* cell suspension (*C. acnes* [AC]). After incubation for 3 h at 37˚C, the silkworm hemolymph was collected. Photographs of the silkworm hemolymph were taken (**A**) and absorbance at 490 nm (A$_{490}$) (**B**) was measured. n = 5/group. Statistically significant differences between groups were evaluated using the Tukey test. *$P < 0.05$. (**C**, **D**) Dose–dependent induction of silkworm hemolymph melanization by autoclaved *C. acnes* cell suspension (*C. acnes* [AC]). Absorbance at 600 nm (A$_{600}$) of the autoclaved *C. acnes* cell suspension (*C. acnes* [AC]) was measured. Silkworms were injected with 50 μl of the autoclaved *C. acnes* cell suspension (*C. acnes* [AC]; A$_{600}$ = 0.625–5). After incubation for 3 h at 37˚C, the silkworm hemolymph was collected. Photographs of the silkworm hemolymph were taken (**C**), and absorbance at 490 nm (A490) (**D**) was measured. The arbitrary unit was defined as the absorbance at 490 nm of a sample of silkworm hemolymph (50 μl) mixed with saline (50 μl).

and/or lysozyme (Fig 6A). Silkworm hemolymph melanization activity induced by the supernatant and precipitate fractions obtained after treating with both lysostaphin and lysozyme was lower than that induced by the BDppt (Fig 6). On the other hand, treatment with each enzyme alone tended to decrease the melanization-inducing ability in the silkworm hemolymph, but the difference was not statistically significant (Fig 6). Moreover, injection of the supernatant fractions obtained after treatment with lysostaphin and/or lysozyme did not induce melanization of the silkworm hemolymph (Fig 6A, 6F, and 6G). These findings suggest that a water-insoluble active substance of *C. acnes* is sensitive to peptidoglycan-degrading enzymes and that degraded water-soluble peptidoglycans did not have melanization inducing activity.

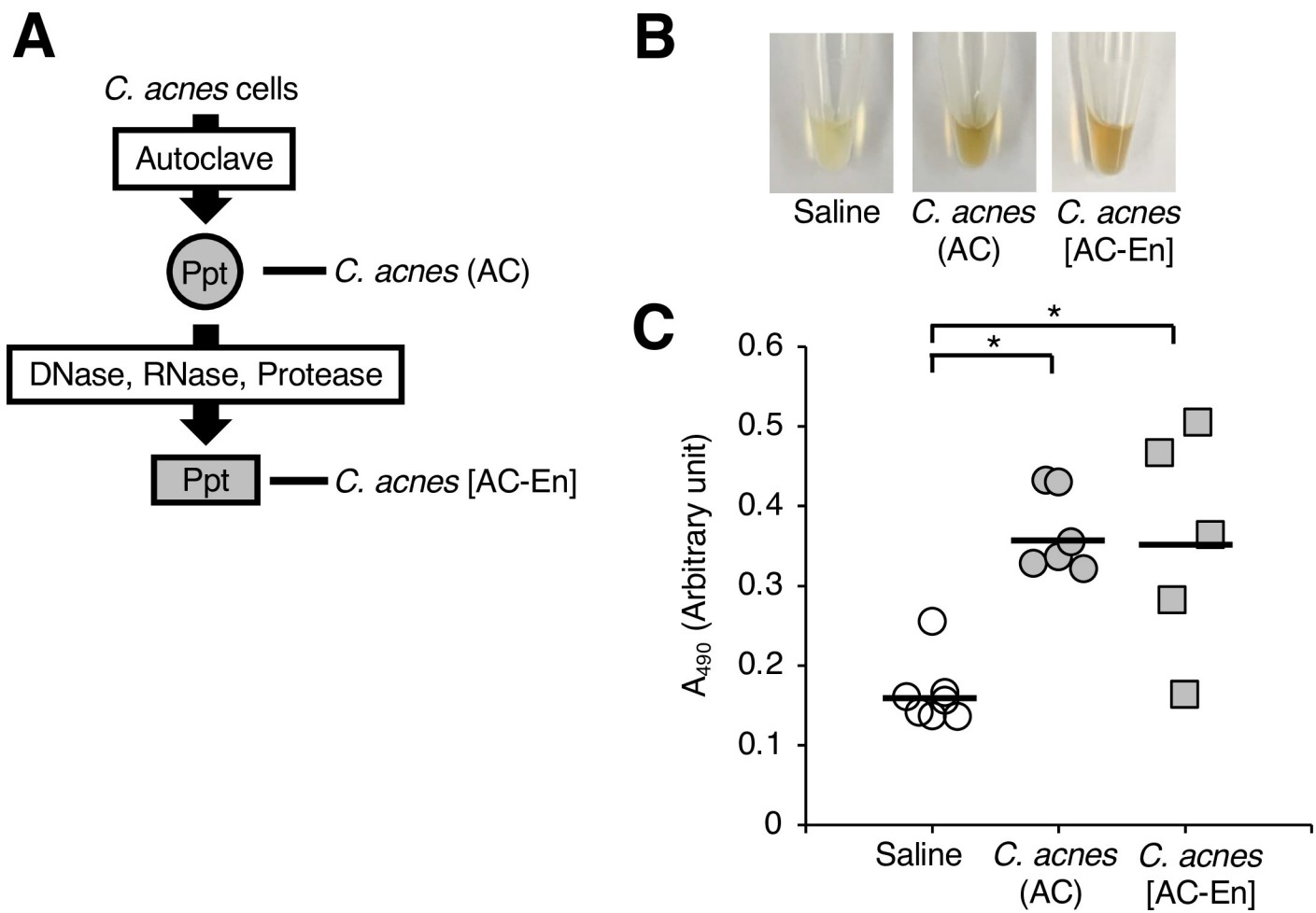

**Fig 4. Effect of various enzyme treatments against the heat–killed *C. acnes* cells on their ability to induce melanization of the silkworm hemolymph.** (**A**) Preparation of the enzyme–treated *C. acnes* (AC) fraction (*C. acnes* [Ac–En]). The *C. acnes* (AC) fraction was treated with DNase, RNase A, and protease K. (**B**, **C**) Silkworms were injected with 50 µl of saline (Saline), the autoclaved *C. acnes* cell suspension (*C. acnes* [AC]), or the enzyme–treated *C. acnes* (AC) fraction (*C. acnes* [Ac–En]). After incubation for 3 h at 37˚C, the silkworm hemolymph was collected. Photographs of the silkworm hemolymph were taken (**B**), and absorbance at 490 nm ($A_{490}$) (**C**) was measured. The arbitrary unit was defined as the absorbance at 490 nm of a sample of silkworm hemolymph (50 µl) mixed with saline (50 µl). n = 5–7/group. Statistically significant differences between groups were evaluated using the Tukey–Kramer test. *$P < 0.05$.

## Discussion

The results of the present study demonstrated that injection of *C. acnes* cells into silkworms induces melanization, a darkening of the silkworm hemolymph, within 3 h, whereas injection of *S. aureus* does not induce rapid silkworm hemolymph melanization. Various fractionations and enzymatic treatments revealed that water-insoluble peptidoglycans of *C. acnes* induce melanization in the silkworm hemolymph. These findings suggest that *C. acnes* induces silkworm innate immunity more rapidly than *S. aureus* and that the water-insoluble peptidoglycans of *C. acnes* are the active melanization-inducing substance.

Since we established a silkworm infection model with *C. acnes* under rearing conditions of 37˚C, which corresponds to human body temperature [31], the rearing condition at 37˚C was used in this study. On the other hand, the standard silkworm rearing temperature is 25–27˚C. We confirmed that the melanization of silkworm hemolymph induced by *C. acnes* in the rearing condition at 37˚C was similar to that at 27˚C (S1 Fig in S1 File). The result suggests that

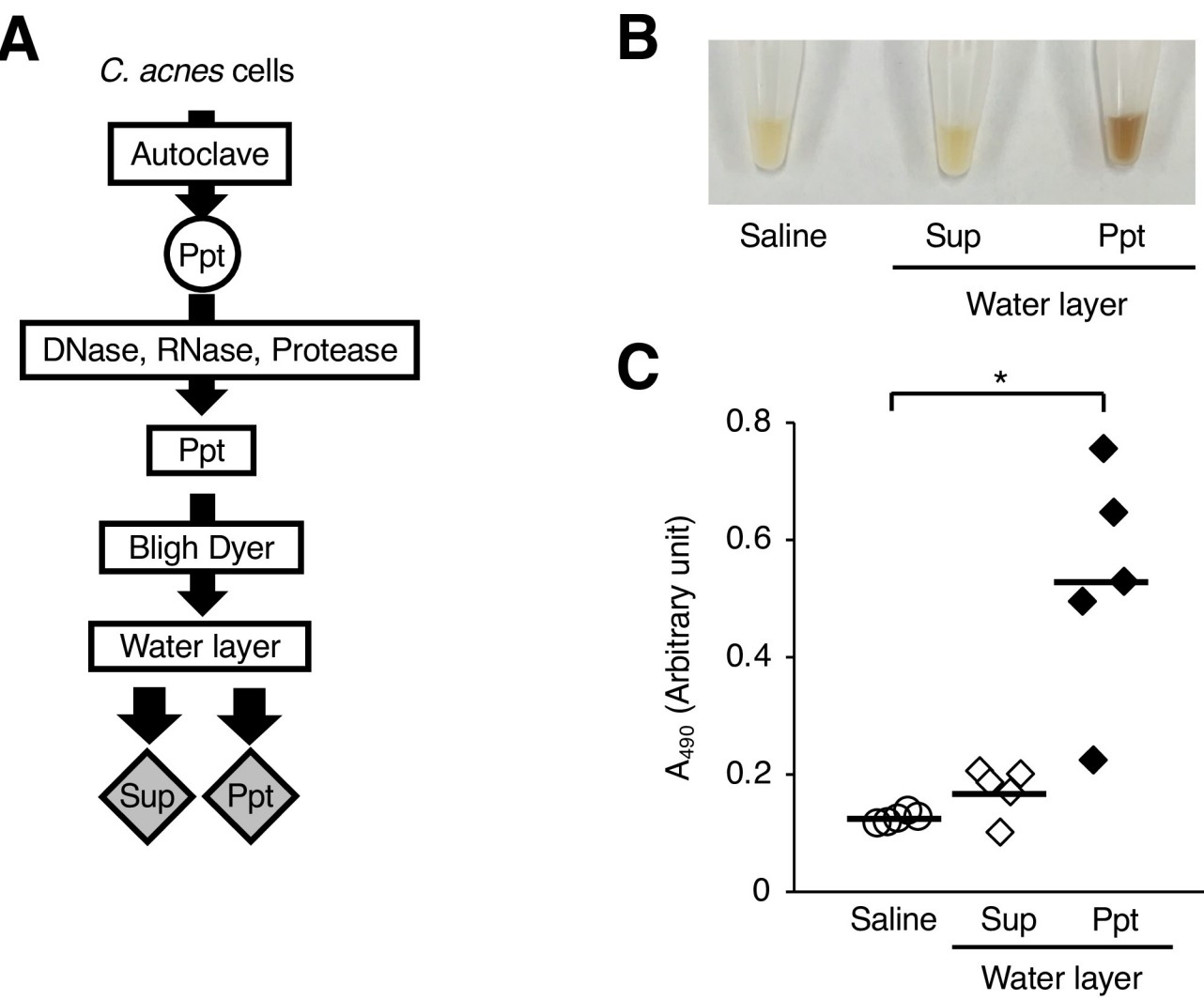

**Fig 5. Induction of silkworm hemolymph melanization by injection of the water–insoluble *C. acnes* fraction obtained following the Bligh–Dyer procedure.** (**A**) Preparation of the aqueous fraction and the chloroform–methanol fraction by the Bligh–Dyer method from the enzyme–treated *C. acnes* (AC) fraction (*C. acnes* [Ac–En]). The water–soluble (Sup) and water–insoluble (Ppt) fractions were obtained. (**B**, **C**) Silkworms were injected with 50 μl of saline (Saline), the water–soluble fraction (Sup), or the water–insoluble fraction (Ppt). After incubation for 3 h at 37˚C, the silkworm hemolymph was collected. Photographs of the silkworm hemolymph were taken (**B**) and absorbance at 490 nm (A490) (**C**) was measured. The arbitrary unit was defined as the absorbance at 490 nm of a sample of silkworm hemolymph (50 μl) mixed with saline (50 μl). n = 5/group. Statistically significant differences between groups were evaluated using the Tukey test. $^{*}P < 0.05$.

the phenoloxidase has activity at 27˚C and 37˚C *in vivo*. We also confirmed that the protein concentration of silkworm hemolymph did not alter by melanization after injection of *C. acnes* sample (S2 Fig in S1 File). The melanization is caused by several steps including ligand recognition, protease cascade, and maturation of phenoloxidase [26, 28]. Therefore, the determination of the kinetic parameters of the melanization reaction is difficult in an *in vivo* experiment. Ligand recognition is the first step of the melanization reaction [26, 28]. The determining kinetic parameters of ligand recognition using purified ligand and receptor *in vitro* is an important future subject.

Melanization of the silkworm hemolymph occurs through oxidative reactions caused by the recognition of microorganism components such as peptidoglycans and β-1,3-glucans [26]. Inflammation such as acne vulgaris is caused by excess activation of the innate immune

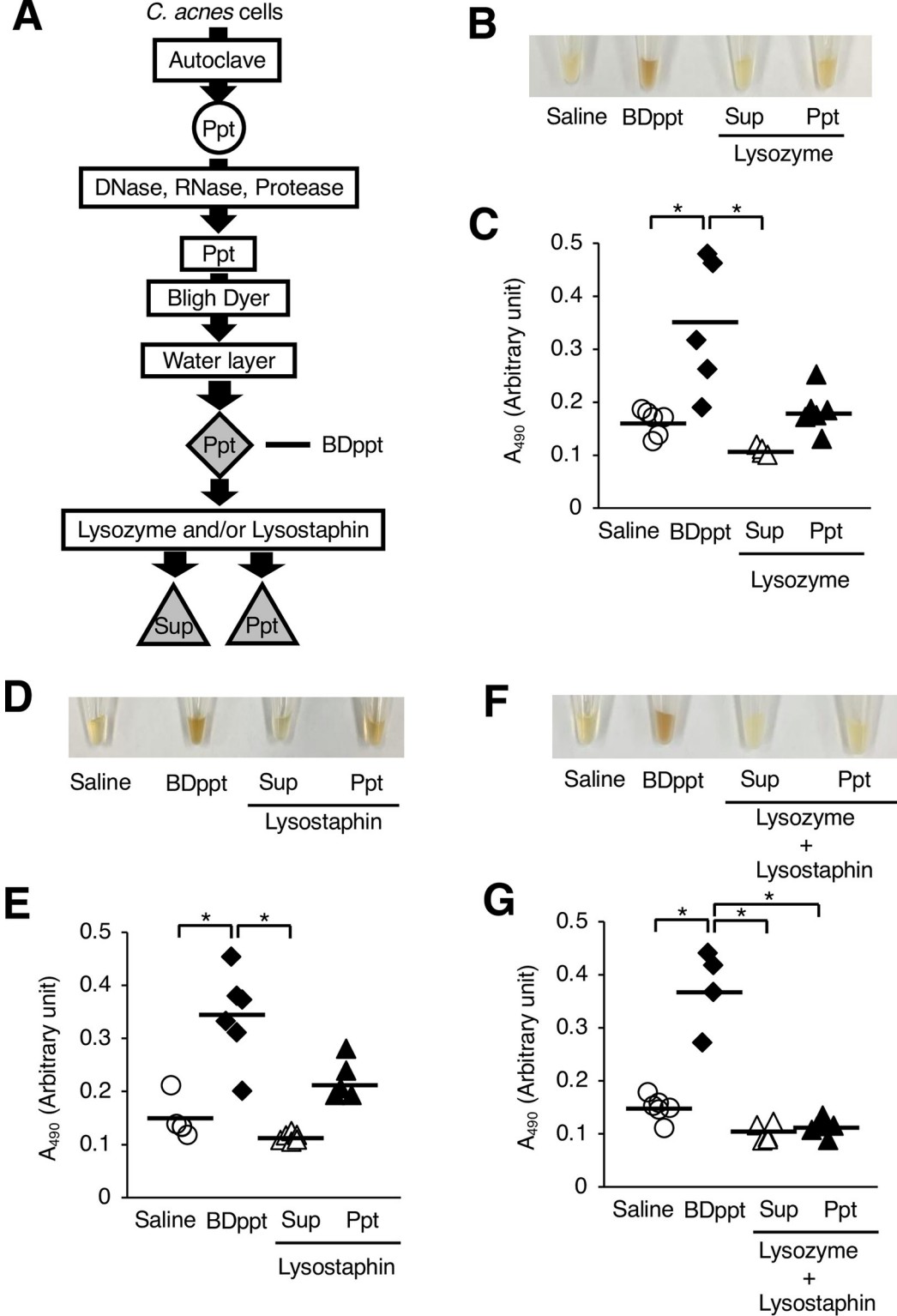

**Fig 6. Reduction of melanization–inducing activity of the water–insoluble *C. acnes* fraction by treatment with peptidoglycan–degrading enzymes.** (**A**) Preparation of the peptidoglycan–degrading enzyme–treated fractions. The water–insoluble fraction obtained following the Bligh–Dyer procedure was called the BDppt fraction. The BDppt fraction was treated with lysostaphin and/or lysozyme. Silkworms were injected with 50 μl of saline (Saline), the BDppt fraction (BDppt), or the peptidoglycan–degrading enzyme–treated fractions. After incubation for 3 h at 37°C, the silkworm

hemolymph was collected. Photographs of the silkworm hemolymph were taken (**B**, **D**, **F**) and absorbance at 490 nm (A$_{490}$) (**C**, **E**, **G**) was measured. (**B**, **C**) Lysozyme treatment experiment. (**D**, **E**) Lysostaphin treatment experiment. (**F**, **G**) Lysozyme and lysostaphin treatment experiment. The arbitrary unit was defined as the absorbance at 490 nm of a sample of silkworm hemolymph (50 μl) mixed with saline (50 μl). n = 4–6/group. Statistically significant differences between groups were evaluated using the Tukey–Kramer test. *$P < 0.05$.

response mediated by pattern recognition receptors, such as Toll-like receptors [11]. Melanization and Toll pathway-mediated immune responses in the silkworm mediate the same signaling pathway [30]. *Bombyx mori* peptidoglycan recognition proteins S1, S4, S5, and L6 in the silkworm hemolymph bind to *S. aureus* peptidoglycans [34]. Moreover, *Bombyx mori* peptidoglycan recognition proteins S1, S4, and S5 enhance melanization via binding to *S. aureus* peptidoglycans [34]. On the other hand, Bmintegrin β3, an integrin expressed in silkworm blood cells, binds to *S. aureus* and inhibits the melanization reaction [35]. In this study, *C. acnes* cells induced melanization of the silkworm hemolymph whereas *S. aureus* cells did not, suggesting that the melanization inhibitory factors like Bmintegrin β3 recognize *S. aureus* cells, but not *C. acnes* cells. Moreover, purified *S. aureus* peptidoglycans induced melanization of the silkworm hemolymph (S3 Fig in S1 File). We assumed that the Bmintegrin β3 has higher inhibitory activity, which suppresses acute melanization by *S. aureus* peptidoglycans. The effects of *C. acnes* cells on melanization inhibitory factors will be an important topic for future studies.

Even *C. acnes* cells killed by autoclaving induce melanization of the silkworm hemolymph. Therefore, the survival of *C. acnes* may not play a significant role in melanization caused by *C. acnes*, and heat-tolerant substances are involved. The melanization-inducing active substance of *C. acnes* was sensitive to peptidoglycan-degrading enzymes, suggesting that water-insoluble peptidoglycans are an active component. These findings suggest that the silkworm can be used to evaluate the ability of *C. acnes* peptidoglycans to induce innate immune reactions.

Peptidoglycans are polymers composed of glycopeptides containing N-acetylglucosamine and N-acetylmuraminic acid [36]. Peptide subunits comprising several amino acids bind to the carboxyl group of N-acetylmuraminic acid [37]. The types of amino acids involved in the peptide subunits and cross-linked structures differ among gram-positive bacteria [37]. Moreover, the sites of amino acid binding in the cross-linked structures differ [37]. The *S. aureus* peptidoglycans consist of a peptide unit composed of L-Ala, D-Glu, L-Lys, and D-Ala bound to muramic acid by a pentaglycine bridge with a 3–4 cross-link [38]. On the other hand, the peptidoglycans of *Corynebacterium pointsettiae*, another gram-positive bacterium, comprise peptide units consisting of Gly D-Glu, L-Hse (homoserine), and D-Ala bound to muramic acid by D-Orn bridge at a 2–4 cross-link [37]. These structural differences in peptidoglycans may affect the robustness of the cell wall and its ability to induce innate immunity. *S. aureus* and *B. subtilis* peptidoglycans induced the melanization of silkworm hemolymph, but *M. luteus* peptidoglycan did not (S3 Fig in S1 File). The result suggests that the activities of peptidoglycans in inducing the melanization of the silkworm hemolymph were different among species in the *in vivo* assay system. We assume that the differences in the melanization-inducing ability between *C. acnes* and *S. aureus* may be due to differences in their peptidoglycan structures. Lysozyme is an enzyme that cleaves the β-1,4 linkage between N-acetylglucosamine and N-acetylmuramic acid in peptidoglycans [39, 40]. Lysostaphin is an enzyme that cleaves the pentaglycine crosslinker in *S. aureus* peptidoglycans [41]. The sites in *C. acnes* peptidoglycans cleaved by these enzymes might be important for recognition by molecules that induce melanization of the silkworm hemolymph. Further studies are required to determine the structures of the water-insoluble *C. acnes* peptidoglycans.

Since the melanization experiments using an individual silkworm are needed to inject the suspension samples into silkworms, individual differences occur. A sample, which precipitates

quickly in a syringe, causes the differences. The optimization of the experimental condition for the precipitate samples is an important subject.

## Conclusion

Silkworms can be used to evaluate the innate immune activation of *C. acnes* and the water-insoluble peptidoglycans of *C. acnes* are important for its hemolymph melanization activity. The silkworm innate immune system is expected to be useful for evaluating compounds that affect innate immune responses caused by *C. acnes*, such as acne vulgaris and intervertebral disc inflammation.

## Supporting information

**S1 File.**
(DOCX)

**S1 Dataset. Datasets included in this study.**
(XLSX)

## Acknowledgments

We thank Tae Nagamachi, Asami Yoshikawa, Yu Sugiyama, Sachi Koganesawa, and Hiromi Kanai (Meiji Pharmaceutical University) for technical assistance rearing the silkworms. We thank Yuki Tateyama (Meiji Pharmaceutical University) for technical assistance supporting the melanization assay.

## Author Contributions

**Conceptualization:** Yasuhiko Matsumoto.

**Data curation:** Yasuhiko Matsumoto, Eri Sato.

**Formal analysis:** Yasuhiko Matsumoto, Eri Sato.

**Funding acquisition:** Yasuhiko Matsumoto.

**Investigation:** Yasuhiko Matsumoto, Eri Sato.

**Project administration:** Yasuhiko Matsumoto.

**Resources:** Takashi Sugita.

**Supervision:** Yasuhiko Matsumoto, Takashi Sugita.

**Validation:** Yasuhiko Matsumoto.

**Visualization:** Yasuhiko Matsumoto, Eri Sato.

**Writing – original draft:** Yasuhiko Matsumoto.

**Writing – review & editing:** Yasuhiko Matsumoto, Eri Sato, Takashi Sugita.

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
