## [Decision Letter · Decision Letter 0]

20 Jul 2022

PONE-D-22-18420Acute innate immune activation in silkworm by the human commensal bacterium Cutibacterium acnesPLOS ONE

Dear Dr. Matsumoto,

Thank you for submitting your manuscript to PLOS ONE. After careful consideration, we feel that it has merit but does not fully meet PLOS ONE’s publication criteria as it currently stands. Therefore, we invite you to submit a revised version of the manuscript that addresses the points raised during the review process.

ACADEMIC EDITOR: 

Editor comments. This is a limited study and basically a confirmative approach showing that melanization is increased in rate following incubation with silkworm hemolymph and bacteria. A few experiments are performed trying to find the responsible agent for activation but this has been done more carefully several times before in several different insect species and crustaceans and therefore it is a requirement for the authors to read up on this area and cite appropriate papers and reviews.

 No information about hemolymph protein amount is given and that has to be included. Further the values in Figure 1 need to be extended to include at least three more time points. What is absorbance in “arbitrary units” on the y-axis. This should be explained.

How do we know that 37C is the optimum temperature for this phenoloxidase? This has to be shown.

In addition how do we know that the enzyme reactions are determined at proper kinetics? This has to be detailed!

In Figure 3D more points (experiments) need to be performed and included. As now is,  only two concentrations are given. 

The values on the x-axis is this some sort of concentration for the bacteria suspension?.If so give detailed information in Figure legend.

Why are bacterial cells treated with DNase and RNase? .

In addition to my comments all concerns raised by the two reviewers must be responded to and the manuscript revised accordingly

We look forward to receiving your revised manuscript.

Kind regards,

Kenneth Söderhäll

Academic Editor

PLOS ONE

Journal Requirements:

Reviewers' comments:

Reviewer's Responses to Questions

**Comments to the Author**

1. Is the manuscript technically sound, and do the data support the conclusions?

Reviewer #1: Partly

Reviewer #2: No

2. Has the statistical analysis been performed appropriately and rigorously? 

Reviewer #1: N/A

Reviewer #2: No

3. Have the authors made all data underlying the findings in their manuscript fully available?

Reviewer #1: Yes

Reviewer #2: No

4. Is the manuscript presented in an intelligible fashion and written in standard English?

Reviewer #1: No

Reviewer #2: Yes

5. Review Comments to the Author

Reviewer #1: The manuscript reports “Acute innate immune activation in silkworm by the human commensal bacterium Cutibacterium acnes”. Although the manuscript contains results from several experiments, I found not much new informative data relevant to the roles of Cutibacterium acnes in silkworm infection model (for bacterial detection) and also about the involvement of that in acute innate immune activation in silkworm. Furthermore, the manuscript is not well organized. There are several major problems. The objective of the research is not clear. It cannot be accepted for publication as it is and required further clarification of the work and in the revision. Please consider the following comments and suggestions:

: The manuscript is not well presented and there are major problems with some unclear results.

: The title is too broad and not suitable to the contents of this manuscript.

: Activation of PO activity in silkworm and effect by C. acnes need better control. Why they use Staphylococcus aureus to be as a comparison? Why did the authors not try PGNs from different sources of bacteria? More gram-positive bacteria and purified PGNs should be tested.

: The sensitivity and specificity of this bacteria (and/or purified PGN) in the activation of silkworm melanization should be tested.

: For the PO activity experiment, how do authors estimate concentration (of hemolymph protein and pgn?)

Reviewer #2: The authors of this manuscript propose a method for detecting Gram-negative bacteria by applying the melanization reaction of silkworm hemolymph. Although the experimental data in this paper are interesting, as specifically commented below, there is a large variation in the measured melanization reaction data and the reliability of each measurement seems to be problematic.

1) Lines 141-142. There are only two measurement points in the graph in Fig. 1, including 1-hour and 3-hours except for 0-time. Therefore, it is not clear at what point in time the melanization reaction reached its maximum value. The authors of this manuscript should add at least 0.5- and 2- hour measurement points to estimate the time of maximum value.

2) Lines 162-164. In Fig.3B, the number of samples (n) in the experiment is set to 5, but there is a lot of variation in the difference in the melanization reaction values between the samples. The error in the reaction values of the non-heat treatment samples is particularly large. Therefore, it is difficult to determine whether the data difference between the non-thermal treatment samples and the thermal treatment ones is significant or not. It would be better to modify the measurement system (sample volume, incubation time or both?) to reduce the experimental error.

3) Lines 174-186. In Fig. 4C, the error in the reaction values of samples of the [AC-En] fraction is particularly large, and it is difficult to determine whether the data difference between C. acnes [AC] samples and C. acnes [AC-En] ones is significant or not, just like in the case of Fig. 3B mentioned above. In addition, the error ranges of the measurements obtained from the water-insoluble fraction (Ppt) (Fig, 5C) and BDppt (Fig. 6 C and E) are too large.

6. PLOS authors have the option to publish the peer review history of their article (what does this mean?). If published, this will include your full peer review and any attached files.

Reviewer #1: No

Reviewer #2: No

---

## [Author Response · Author response to Decision Letter 0]

6 Sep 2022

ACADEMIC EDITOR: 

Editor comments. This is a limited study and basically a confirmative approach showing that melanization is increased in rate following incubation with silkworm hemolymph and bacteria. A few experiments are performed trying to find the responsible agent for activation but this has been done more carefully several times before in several different insect species and crustaceans and therefore it is a requirement for the authors to read up on this area and cite appropriate papers and reviews.

According to the editor’s comment, we described other research in the Introduction section of the revised manuscript (Page 5, lines 55-57, lines 59-62).

[Page 5, lines 55-57]

In Drosophila melanogaster, hemolymph melanization caused by the recognition of a foreign invader such as bacteria or fungi in the body contributes to coagulating the invader and repairing the wound [25,26].

[Page 5, lines 59-62]

Hemolymph melanization was caused by phenoloxidases activated by serine proteases [25,26]. The phenoloxidases (PPO1 and PPO2)-deficient flies are sensitive to Staphylococcus aureus infection [28]. Therefore, hemolymph melanization in insects can be used as an indicator of innate immune system activation [17,19,29].

No information about hemolymph protein amount is given and that has to be included. 

According to the editor’s comment, we determined the protein concentration of silkworm hemolymph (Supplementary Figure 2). The protein concentration of silkworm hemolymph with melanization after injection of C. acnes (AC) was similar to that of silkworm hemolymph without melanization after injection of saline (Supplementary Figure 2). The result suggests that the protein concentration of silkworm hemolymph did not alter by melanization after injection of C. acnes (AC) sample. We added the sentence in the Discussion section of the revised manuscript (Page 18, lines 272-274).

[Page 18, lines 272-274]

We also confirmed that the protein concentration of silkworm hemolymph did not alter by melanization after injection of C. acnes sample (S2 Fig in S1 File).

In this study, we determined in vivo melanization of silkworm hemolymph using silkworm body, not in vitro melanization using isolated hemolymph proteins. We added the figure of an experimental scheme of this study in Figure 1A of the revised manuscript.

Further the values in Figure 1 need to be extended to include at least three more time points. 

According to the editor’s comment, we performed the time course experiment for 3 h with more time points (0, 0.5, 1, 2, and 3 h) (Figure 1B). The melanization of silkworm hemolymph was increased at 0.5 h after injection of C. acnes cells (Figure 1B). 

What is absorbance in “arbitrary units” on the y-axis. This should be explained.

According to the editor’s comment, we added the explanation of the arbitrary unit in the Materials & Methods section and Figure legends of the revised manuscript (Page 8, lines 105-106, page 11, 157-159, page 12, lines 166-167, page 13, lines 190-191, page 15, lines 228-230, page 16, lines 241-242, page 17, lines 254-255).

The arbitrary unit was defined as the absorbance at 490 nm of a sample of silkworm hemolymph (50 µl) mixed with saline (50 µl). 

How do we know that 37C is the optimum temperature for this phenoloxidase? This has to be shown.

According to the editor’s comment, we performed a new experiment for the effect of temperature on the melanization of silkworm hemolymph induced by C. acnes. The melanization of silkworm hemolymph induced by C. acnes in the rearing condition at 37˚C was similar to that at 27˚C (Supplementary Figure 1). The result suggests that the phenoloxidase has activity at 27˚C and 37˚C in vivo. The description was added to the Discussion section of the revised manuscript (Page 18, lines 267-272).

[Page 18, lines 267-272]

Since we established a silkworm infection model with C. acnes under rearing conditions of 37˚C, which corresponds to human body temperature [31], the rearing condition at 37˚C was used in this study. On the other hand, the standard silkworm rearing temperature is 25-27˚C. We confirmed that the melanization of silkworm hemolymph induced by C. acnes in the rearing condition at 37˚C was similar to that at 27˚C (S1 Fig in S1 File). The result suggests that the phenoloxidase has activity at 27˚C and 37˚C in vivo.

In addition how do we know that the enzyme reactions are determined at proper kinetics? This has to be detailed!

In this study, we detected melanization of the silkworm hemolymph using the silkworm body. The melanization is caused by several steps including ligand recognition, protease cascade, and maturation of phenoloxidase. Therefore, the determination of the kinetic parameters of the melanization reaction is difficult in an in vivo experiment. Ligand recognition is the first step of the melanization reaction. The determining kinetic parameters of ligand recognition using purified ligand and receptor in vitro is an important future subject. The description was added in the Discussion section of the revised manuscript (Page 18-19, lines 274-279).

[Page 18, lines 267-272]

The melanization is caused by several steps including ligand recognition, protease cascade, and maturation of phenoloxidase [26,28]. Therefore, the determination of the kinetic parameters of the melanization reaction is difficult in an in vivo experiment. Ligand recognition is the first step of the melanization reaction [26,28]. The determining kinetic parameters of ligand recognition using purified ligand and receptor in vitro is an important future subject.

In Figure 3D more points (experiments) need to be performed and included. As now is, only two concentrations are given. 

According to the editor’s comment, we performed a new experiment that is dose dependency of the melanization induced by C. acnes (AC). The new data was shown in the Figure 3D of the revised manuscript.

The values on the x-axis is this some sort of concentration for the bacteria suspension?.If so give detailed information in Figure legend.

In Figure 3D, the bacterial suspension was used. According to the editor’s comment, we added the sentence in the Figure legends of the revised manuscript (Page 13, lines 180-181, lines 185-187).

Why are bacterial cells treated with DNase and RNase? .

In the present study, autoclaved bacteria sample included residual DNA and RNA, enzymatic treatment was used to remove it. Research in Drosophila has reported that STING protein is involved in the innate immune response to viral infection (Goto A., et al., Immunity, 2018, 49:225-234). The STING protein activates immunity when pathogen-derived DNA is recognized in Drosophila. Therefore, we want to remove nucleic acids.

Reviewer #1: The manuscript reports “Acute innate immune activation in silkworm by the human commensal bacterium Cutibacterium acnes”. Although the manuscript contains results from several experiments, I found not much new informative data relevant to the roles of Cutibacterium acnes in silkworm infection model (for bacterial detection) and also about the involvement of that in acute innate immune activation in silkworm. Furthermore, the manuscript is not well organized. There are several major problems. The objective of the research is not clear. It cannot be accepted for publication as it is and required further clarification of the work and in the revision. Please consider the following comments and suggestions:

: The manuscript is not well presented and there are major problems with some unclear results.

According to the editor’s comment, we described other research in the Introduction section of the revised manuscript (Page 5, lines 55-57, lines 59-62).

[Page 5, lines 55-57]

In Drosophila melanogaster, hemolymph melanization caused by the recognition of a foreign invader such as bacteria or fungi in the body contributes to coagulating the invader and repairing the wound [25,26].

[Page 5, lines 59-62]

Hemolymph melanization was caused by phenoloxidases activated by serine proteases [25,26]. The phenoloxidases (PPO1 and PPO2)-deficient flies are sensitive to Staphylococcus aureus infection [28]. Therefore, hemolymph melanization in insects can be used as an indicator of innate immune system activation [17,19,29].

: The title is too broad and not suitable to the contents of this manuscript.

According to the reviewer’s comment, the title was changed in the revised manuscript.

Title: Acute melanization of silkworm hemolymph by peptidoglycans of the human commensal bacterium Cutibacterium acnes

: Activation of PO activity in silkworm and effect by C. acnes need better control. Why they use Staphylococcus aureus to be as a comparison? Why did the authors not try PGNs from different sources of bacteria? More gram-positive bacteria and purified PGNs should be tested.

Staphylococcus aureus, like C. acnes, is a gram-positive bacterium present in human skin and induces inflammatory skin diseases such as atopic dermatitis. Therefore, we focused on S. aureus for comparison.

Following the reviewer’s comment, we performed a new experiment that the induction activities of purified S. aureus peptidoglycans on the melanization of silkworm hemolymph. S. aureus peptidoglycans induced the melanization of silkworm hemolymph (Supplementary Figure 3). Therefore, we added the sentences in the Discussion section of the revised manuscript (Page 19, lines 287-294). 

Page 19, lines 287-294

On the other hand, Bmintegrin β3, an integrin expressed in silkworm blood cells, binds to S. aureus and inhibits the melanization reaction [35]. In this study, C. acnes cells induced melanization of the silkworm hemolymph whereas S. aureus cells did not, suggesting that the melanization inhibitory factors like Bmintegrin β3 recognize S. aureus cells, but not C. acnes cells. Moreover, purified S. aureus peptidoglycans induced melanization of the silkworm hemolymph (S3 Fig in S1 File). We assumed that the Bmintegrin β3 has higher inhibitory activity, which suppresses acute melanization by S. aureus peptidoglycans.

: The sensitivity and specificity of this bacteria (and/or purified PGN) in the activation of silkworm melanization should be tested.

Following the reviewer’s comment, we performed a new experiment that the induction activities of purified gram-positive bacterial peptidoglycan on the melanization of silkworm hemolymph. S. aureus and B. subtilis peptidoglycans induced the melanization of silkworm hemolymph, but M. luteus peptidoglycan did not (Supplementary Figure 3). The result suggests that the activities of peptidoglycans in inducing the melanization of the silkworm hemolymph were different among species in the in vivo assay system. We assumed that the differences in the structure of peptidoglycan among species affect the binding to recognition receptors such as PGRPs, peptidoglycan recognition proteins. The description was added in the Discussion section of the revised manuscript (Pages 20-21, lines 308-312).

Pages 20-21, lines 312-315.

S. aureus and B. subtilis peptidoglycans induced the melanization of silkworm hemolymph, but M. luteus peptidoglycan did not (S3 Fig in S1 File). The result suggests that the activities of peptidoglycans in inducing the melanization of the silkworm hemolymph were different among species in the in vivo assay system.

: For the PO activity experiment, how do authors estimate concentration (of hemolymph protein and pgn?)

According to the reviewer’s comment, we determined the protein concentration of silkworm hemolymph (Supplementary Figure 2). The protein concentration of silkworm hemolymph with melanization after injection of C. acnes (AC) was similar to that of silkworm hemolymph without melanization after injection of saline (Supplementary Figure 2). The result suggests that the protein concentration of silkworm hemolymph did not alter by melanization after injection of C. acnes (AC) sample. We added the sentences in the Discussion section of the revised manuscript (Page 18, lines 272-274).

[Page 18, lines 272-274]

We also confirmed that the protein concentration of silkworm hemolymph did not alter by melanization after injection of C. acnes sample (S2 Fig in S1 File).

In this study, we determined in vivo melanization of silkworm hemolymph using silkworm body, not in vitro melanization using isolated hemolymph proteins. We added the figure of an experimental scheme of this study in Figure 1A of the revised manuscript.

Reviewer #2: The authors of this manuscript propose a method for detecting Gram-negative bacteria by applying the melanization reaction of silkworm hemolymph. Although the experimental data in this paper are interesting, as specifically commented below, there is a large variation in the measured melanization reaction data and the reliability of each measurement seems to be problematic.

1) Lines 141-142. There are only two measurement points in the graph in Fig. 1, including 1-hour and 3-hours except for 0-time. Therefore, it is not clear at what point in time the melanization reaction reached its maximum value. The authors of this manuscript should add at least 0.5- and 2- hour measurement points to estimate the time of maximum value.

According to the editor’s comment, we performed the time course experiment for 3 h with more time points (0, 0.5, 1, 2, and 3 h) (Figure 1B). The melanization of silkworm hemolymph was increased at 0.5 h after injection of C. acnes cells (Figure 1B). 

2) Lines 162-164. In Fig.3B, the number of samples (n) in the experiment is set to 5, but there is a lot of variation in the difference in the melanization reaction values between the samples. The error in the reaction values of the non-heat treatment samples is particularly large. Therefore, it is difficult to determine whether the data difference between the non-thermal treatment samples and the thermal treatment ones is significant or not. It would be better to modify the measurement system (sample volume, incubation time or both?) to reduce the experimental error.

Since the melanization experiments using an individual silkworm are needed to inject the suspension samples into silkworms, individual differences occur. A sample, which precipitates quickly in a syringe, causes the differences. The optimization of the experimental condition for the precipitate samples is an important subject. We added the sentences in the Discussion section of the revised manuscript (Page 21, lines 324-327).

3) Lines 174-186. In Fig. 4C, the error in the reaction values of samples of the [AC-En] fraction is particularly large, and it is difficult to determine whether the data difference between C. acnes [AC] samples and C. acnes [AC-En] ones is significant or not, just like in the case of Fig. 3B mentioned above. In addition, the error ranges of the measurements obtained from the water-insoluble fraction (Ppt) (Fig, 5C) and BDppt (Fig. 6 C and E) are too large.

We hoped that the partially degraded and soluble peptidoglycan would be active to give more stable results, but the soluble fraction did not have an activity. The optimization of the experimental condition for the precipitate samples is an important subject. We added the sentences in the Discussion section of the revised manuscript (Page 21, lines 324-327).

---

## [Editor Report · Decision Letter 1]

12 Sep 2022

Acute melanization of silkworm hemolymph by peptidoglycans of the human commensal bacterium  Cutibacterium acnes

PONE-D-22-18420R1

Dear Dr. Matsumoto,

We’re pleased to inform you that your manuscript has been judged scientifically suitable for publication and will be formally accepted for publication once it meets all outstanding technical requirements.

Kind regards,

Kenneth Söderhäll

Academic Editor

PLOS ONE
---

## [Editor Report · Acceptance letter]

16 Sep 2022

PONE-D-22-18420R1 

Acute melanization of silkworm hemolymph by peptidoglycans of the human commensal bacterium *Cutibacterium acnes*

Dear Dr. Matsumoto:

I'm pleased to inform you that your manuscript has been deemed suitable for publication in PLOS ONE. Congratulations! Your manuscript is now with our production department. 

Kind regards, 

on behalf of

Dr. Kenneth Söderhäll 

Academic Editor

PLOS ONE